# Continuous ultraviolet to blue-green astrocomb

Yuk Shan Cheng[1], Kamalesh Dadi [1], Toby Mitchell[1], Samantha Thompson [2], Nikolai Piskunov[3], Lewis D. Wright[4], Corin B. E. Gawith [4,5], Richard A. McCracken [1] & Derryck T. Reid [1] ✉

Cosmological and exoplanetary science using transformative telescopes like the ELT will demand precise calibration of astrophysical spectrographs in the blue-green, where stellar absorption lines are most abundant. Astrocombs—lasers providing a broadband sequence of regularly-spaced optical frequencies on a multi-GHz grid—promise an atomically-traceable calibration scale, but their realization in the blue-green is challenging for current infrared-laser-based technology. Here, we introduce a concept achieving a broad, continuous spectrum by combining second-harmonic generation and sum-frequency-mixing in an MgO:PPLN waveguide to generate 390–520 nm light from a 1 GHz Ti:sapphire frequency comb. Using a Fabry-Pérot filter, we extract a 30 GHz sub-comb spanning 392–472 nm, visualizing its thousands of modes on a high-resolution spectrograph. Experimental data and simulations demonstrate how the approach can bridge the spectral gap present in second-harmonic-only conversion. Requiring only ≈100 pJ pulses, our concept establishes a new route to broadband UV-visible generation at GHz repetition rates.

In the field of extreme precision radial velocity (EPRV) measurements, the aim is to push the detection threshold to 10 cm s⁻¹. This 10 cm s⁻¹ goal is set to enable the detection of low-mass, longer-period extra-solar planets (exoplanets), in particular those that are similar to our own Earth. On some of the next-generation instruments, such as ANDES on the ELT[1], the aim is to achieve measurements at the 1–2 cm s⁻¹ level, to enable high precision and high accuracy cosmological observations like the Sandage–Loeb Test[2].

To detect an exoplanet via the radial velocity (RV) method requires the collection of high-resolution spectra of its host star, over a timescale corresponding to at least three orbits of the exoplanet. The shift in wavelength of all the spectral lines is measured (Doppler shift) and recorded over time. For example, for the Earth in orbit around the Sun (viewed edge-on), this radial velocity timeseries would appear as a sinusoid with an amplitude of 9 cm s⁻¹ and a period of 1 year. To put a 10 cm s⁻¹ Doppler shift in perspective, the size of a single pixel on many current high-resolution spectrographs used for RV measurements is of the order of 1 km s⁻¹, illustrating how these measurements require extremely precise wavelength calibration.

Astrocombs—broadband laser frequency combs with multi-GHz spacing—represent the ideal wavelength calibrator[3] for these tasks. To enable the detection of an Earth-analog exoplanet, the RV measurement must be stable over many years, have a single measurement precision of ≈1 m s⁻¹, have spectral features that are regular and sufficiently spaced, unresolved (i.e., narrow intrinsic width), and cover the entire spectral format so that maximum information can be extracted from the stellar spectrum. Laser frequency combs can provide all these features, by virtue of their stability, atomically-referenceable accuracy, sub-100-kHz linewidths, multi-GHz spacing, and potential for broadband spectral coverage[4,5].

[1]Institute of Photonics and Quantum Sciences, Heriot-Watt University, Edinburgh EH14 4AS, UK. [2]Astrophysics Group, Cavendish Laboratory, J.J. Thomson Avenue, Cambridge CB3 0HE, UK. [3]Department of Physics and Astronomy, Uppsala University, Box 516, 751 20 Uppsala, Sweden. [4]Covesion Ltd, Unit F3, Adanac North, Adanac Drive, Nursling, Southampton SO16 0BT, UK. [5]Optoelectronics Research Centre, University of Southampton, Southampton, Hampshire SO17 1BJ, UK. ✉e-mail: D.T.Reid@hw.ac.uk

Radial velocity precision depends on the number of strong spectral features[6], and for solar-like stars, the density of such lines increases towards the blue end of the spectrum. This is why the most advanced RV instruments today, e.g., HARPS (380–690 nm[7]), EXPRES (380–680 nm[8]), and ESPRESSO (378–789 nm[9]), reach out to the blue part of the spectrum. Future science cases such as the Sandage–Loeb test use the cosmological redshifted Lyman-alpha forest, with the wavelengths starting at ≈365 nm. Converting this advantage to RV precision requires accurate and reproducible wavelength calibration in the blue and near-ultraviolet. Alternative calibration methods (principally hollow-cathode lamps) do not provide intrinsic accuracy, homogeneous wavelength coverage, or stability of an LFC[3].

Blue-visible astrocombs have previously been demonstrated using a variety of approaches. The second-harmonic generation of near-infrared mode-locked lasers has produced narrowband astrocombs at 400 nm[10,11]. Supercontinuum generation in photonic-crystal fiber has enabled blue-green (435–600 nm[12]), green (530–560 nm[13]), and green-red (500–620 nm[14]) astrocombs. Recent improvements in the manufacture of silicon nitride (SiN) and periodically-poled lithium niobate (PPLN) waveguides have led to on-chip approaches which exploit $\chi^{(2)}$ and $\chi^{(3)}$ processes. Obrzud et al.[15] demonstrated 400–600 nm of gap-free coverage at 10 GHz mode spacing through $\chi^{(3)}$ triple-sum-frequency-generation of an electro-optic modulator comb in SiN; however the intrinsically wide mode spacing of the pump source required amplification, broadening, and compression stages to achieve sufficient peak power for nonlinear conversion, despite the enhancement offered by waveguide confinement. Nakamura et al.[16] reported a 30 GHz astrocomb which extended down to 350 nm via second, third, and fourth harmonic generation of a 1.5 μm source in a chirped PPLN waveguide, however, wide gaps remained in the spectral coverage, and the 250 MHz pump laser required multiple filtering and amplification stages to achieve the desired mode spacing. A similar approach has recently been demonstrated using an EOM pump source[17].

In this work, we propose and experimentally demonstrate a new approach that achieves a broad, continuous visible spectrum by combining second-harmonic generation (SHG) and sum-frequency-mixing (SFM) in an aperiodically-poled MgO:PPLN waveguide. On its own, SHG suppresses weaker spectral features contained in the fundamental light, since it is a quadratic ($\chi^{(2)}$) nonlinearity, but by using a strong auxiliary pulse, SFM can be used to linearly transfer weak but broad infrared components into the visible. We illustrate this experimentally by using a Ti:sapphire laser frequency comb operating with a repetition frequency of $f_{rep}$ = 1 GHz to generate 390–520 nm light from an infrared supercontinuum, achieving a gap-free frequency bandwidth of >90 THz. The wide mode spacing of this visible comb means that a relatively low finesse Fabry–Pérot cavity can then be used to extract a sub-comb spaced at a high multiple of $f_{rep}$, and providing strong suppression of nearest-neighbor 1-GHz modes. We implement such a scheme by locking a low-dispersion Fabry–Pérot filter cavity to the incident comb modes to produce a 30 GHz astrocomb spanning 392–472 nm and limited only by the mirror reflectivity bandwidth. The resulting comb is visualized on a comb-mode-resolving cross-dispersion echelle-prism spectrograph, demonstrating well-resolved comb lines across all 24 diffraction orders.

## Results

### Visible astrocomb generation concept

While $\chi^{(3)}$-mediated supercontinua are readily available in the infrared, with few-pJ octave-spanning generation being reported[18], equivalent performance in the ultraviolet to the blue-green region is more challenging because of the tightly-confined waveguide structures required to achieve the necessary group-delay dispersion (GDD) profile for efficient, broadband generation. Instead, our approach uses a pump pulse to generate a broad, near-infrared supercontinuum and employs

SHG and SFM between the supercontinuum and a replica pump pulse to create a broad, gap-free visible comb. Since the supercontinuum light has the same carrier-envelope offset frequency ($f_{CEO}$) as the pump, the SHG and SFM components share identical comb offsets of $2f_{CEO}$ and are indistinguishable within the comb. As Fig. 1a illustrates, when the pump and supercontinuum fields are incident independently on a $\chi^{(2)}$ medium they are converted in a way that suppresses weaker supercontinuum components because of the quadratic SHG process. By temporally overlapping the pump and supercontinuum fields, SFM can be used to bridge such gaps, efficiently transferring the supercontinuum bandwidth into the visible region. However, this process alone is not enough to enable a practical implementation, which also requires an efficient means of realizing $\chi^{(2)}$ conversion across an exceptionally broad bandwidth. For this purpose we use an MgO:PPLN ridge waveguide (Fig. 1b)[19], which offers high coupling efficiencies, strong confinement, and broadband conversion in a suitably designed quasi-phasematched grating structure (see Methods). Figure 1d shows the spectra of the modeled fundamental and supercontinuum fields (red shading) and the spectrum of the SHG/SFM field produced by the optimized grating design (blue shading). The delay-resolved spectra shown in Fig. 1e illustrate how the addition of SFM between the pump and supercontinuum fields generates around 100 THz of additional bandwidth having high spectral intensity in the 420–470 nm region.

### Experimental astrocomb preparation and characterization

We employed a Ti:sapphire laser comb delivering 33 fs, 1 GHz pulses at 800 nm (Fig. 2a) to pump a 35-mm-long photonic-crystal fiber (PCF)[20], whose closely spaced zero-dispersion wavelengths of 775 and 945 nm and high nonlinearity ($\gamma \approx 130$ W$^{-1}$ km$^{-1}$) produced a smooth 650–1050 nm spectrum (Fig. 2d). After an appropriate optical delay, replica pump pulses were co-launched with the supercontinuum pulses into an MgO:PPLN waveguide (Fig. 2a, inset). The waveguide was driven by a pump and supercontinuum pulses with energies of 300 and 100 pJ, respectively (Fig. 2d, red shading), and produced several milliwatts of ultraviolet to blue-green output from 390–520 nm (Fig. 2d, blue shading). These spectra broadly match the original design predictions (Fig. 1d), but in practice a weaker infrared supercontinuum component reduced the blue generation above 450 nm. Figure 2b shows the experimentally measured delay-resolved SHG/SFM spectrum, which was qualitatively similar to the a priori design in Fig. 1e. The ultraviolet to blue-green spectrum directly from the waveguide was easily observable visually, as shown in the photograph of Fig. 2c, where the waveguide output was dispersed and imaged on a screen.

An astrocomb with a 30 GHz mode spacing was prepared by coupling the output of the MgO:PPLN waveguide into a Fabry–Pérot etalon composed of two plane-parallel mirrors with ≈99% reflectivity coatings from 390–470 nm, which exhibited very low GDD over the same wavelength range and zero GDD at 430 nm. The 1 GHz mode spacing of the master comb allows good nearest-neighbor comb-mode suppression, while relaxing the requirements on the etalon finesse (≈300), in turn enabling the design of mirror coatings that exhibit a low GDD across a wide bandwidth. The etalon was dither-locked (see Methods) to selectively transmit one sub-comb with a spacing of 30 GHz.

An echelle-prism cross-dispersion spectrograph with an estimated resolving power of ≈46,000 (see Methods) was configured to visualize the resulting comb modes. This measurement, presented in Fig. 3, provides direct confirmation that the combination of SHG and SFM is effective in delivering a gap-free field of high-density comb modes from 392–472 nm. Comb modes are present throughout each echelle order, as illustrated by the insets contained in Fig. 3, which show the comb-mode details in magnified regions of the echellogram selected to represent the full spectral range. Furthermore, the

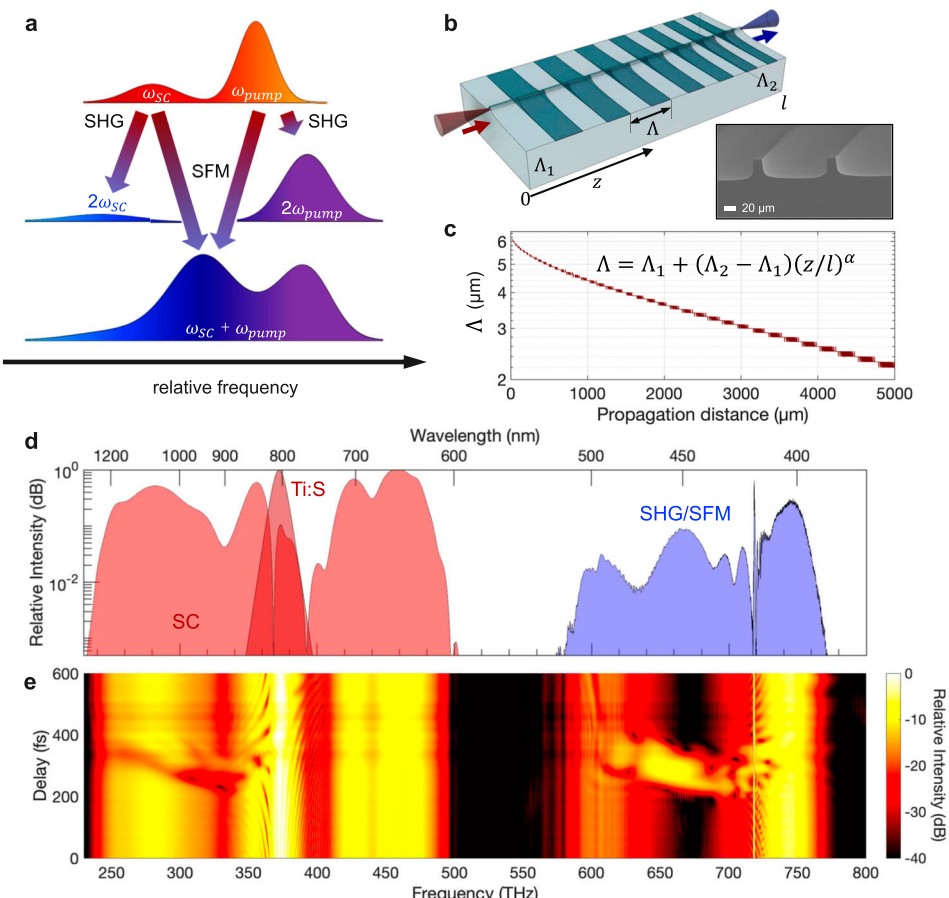

**Fig. 1 | Concept of broadband UV to blue-green generation using sum-frequency mixing. a** The infrared component of a broadband supercontinuum (SC) pulse is mixed in a $\chi^{(2)}$ medium with a higher frequency, stronger pump pulse. In the absence of temporal overlap, only second-harmonic generation (SHG) occurs, with the quadratic nonlinearity suppressing weaker spectral components, and leading to gaps in the generated visible spectrum. By introducing an appropriate delay, allowing the pump and SC pulses to walk through each other, sum-frequency-mixing occurs, which blue-shifts and enhances the supercontinuum components in a way that provides gap-free UV-blue generation. **b** Efficient conversion of 1 GHz repetition-rate, ≈100 pJ pulses is achieved by using an MgO:PPLN

ridge waveguide containing a custom-designed quasi-phasematched grating. Inset: Scanning electron-beam micrograph of a representative waveguide. **c** The MgO:PPLN domain-width pattern follows an aperiodic design optimized to ensure gap-free upconversion. The optimized QPM grating design has entry and exit periods of $\Lambda_1 = 6.3\,\mu m$ and $\Lambda_2 = 2.2\,\mu m$, respectively, and a nonlinearity of $\alpha = 0.48$, corresponding to a structure (**b**) with domain sizes that become progressively smaller towards the end of the waveguide. **d** Simulated 30-fs pump pulse and supercontinuum (red) and resulting SHG and SFM spectrum achieved for optimal pump-supercontinuum delay ($\tau = 250$ fs). **e** Simulated delay-resolved spectra (step size, $\triangle\tau = 10$ fs) illustrating the additional bandwidth created by the SFM process.

astrocomb coverage from 392–472 nm entirely fills the operational range of our Fabry–Pérot etalon (see Methods).

## Discussion

In this work we have introduced a new concept for achieving a gap-free, broadband ultraviolet to blue-green astrocomb, which until now has remained only partially resolved by other technical approaches, which have been unable to offer the necessary combination of mode spacing, bandwidth, and gap-free spectral structure. We have demonstrated an astrocomb spanning 392–472 nm with 30 GHz mode spacing, corresponding to over 4000 individually resolvable comb modes across a frequency bandwidth of 130 THz. The mode format is well matched to the calibration requirements of future high-resolution astronomical spectrographs, such as HARPS3[21] and ANDES[1], which require a comb-mode spacing of 32 GHz for wavelengths in the 370–600 nm UB band[22], corresponding, at the shortest wavelength, to one comb line per three resolution elements for a nominal resolving power of $R > 75{,}000$. Our approach is readily extended to provide complete coverage of the important visible band by combining it with proven Ti:sapphire-based schemes utilizing photonic-crystal fiber to produce visible combs covering 450–900 nm[23,24]. More generally, the

demonstration presented here is an important step towards obtaining a gap-free astrocomb spanning the complete ultraviolet to the near-infrared range, for example, the 400–1800 nm range specified in the European Southern Observatory's technical specification for the ELT-ANDES spectrograph[25]. Ti:sapphire lasers operating at 1 GHz are readily extended to this near-infrared range by using optical parametric oscillators[26], and a broadband near-infrared astrocomb has been reported by spectrally broadening such an OPO source in a highly nonlinear fiber[27].

Our results also extend the understanding of MgO:PPLN ridge waveguide devices for ultraviolet/blue generation[28]. While mechanically fabricated MgO:PPLN waveguides have been shown to be damage resilient and capable of multi-watt average power handling[29], in comparison to emerging thin-film lithium niobate waveguides[30], their larger cross-sectional areas are not intrinsically single-mode for visible wavelengths. Despite this, we have consistently observed a high-quality transverse visible mode in the visible, which has enabled successful coupling into the high finesse Fabry–Pérot etalon used for longitudinal mode selection. We attribute this primarily to the efficient transfer, through the nonlinear conversion process, of the fundamental TE0 pump mode profile into the SHG and SFM fields.

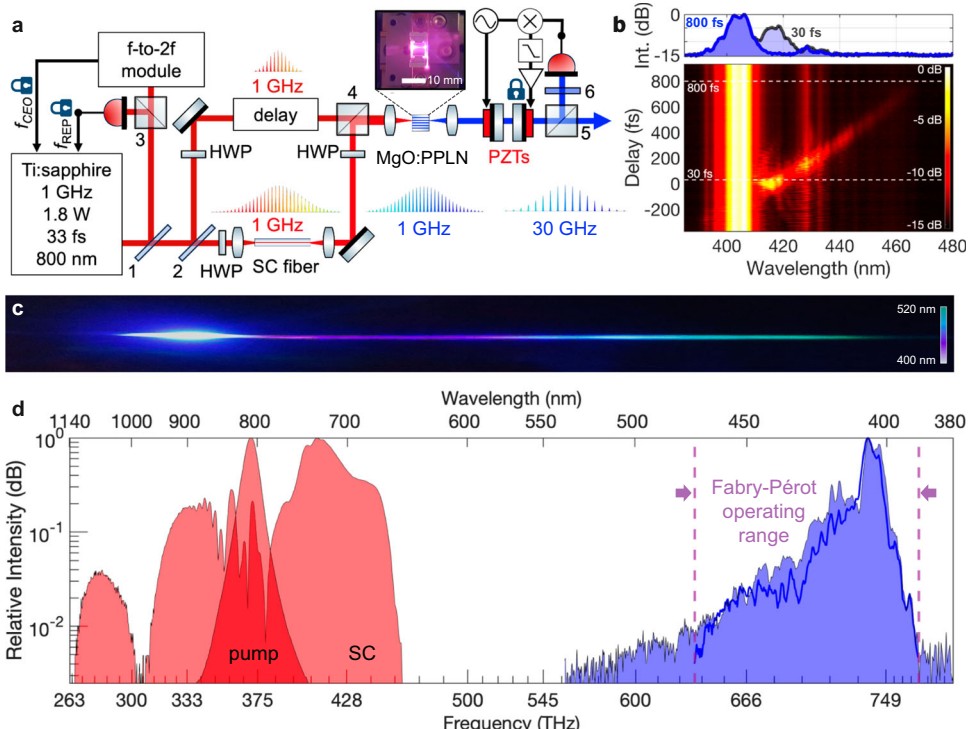

**Fig. 2 | Implementation of broadband UV to blue-green generation. a** Pump pulses from a 1 GHz Ti:sapphire comb enter a 35 mm long photonic-crystal fiber to create 650–1150 nm supercontinuum (SC) pulses, which are launched into an MgO:PPLN waveguide (inset). Replica pump pulses are co-launched into the waveguide and overlapped with infrared SC components to achieve a broadband SHG and SFM comb, which is filtered by a Fabry–Pérot etalon to produce a 30 GHz astrocomb. All beamsplitters are non-polarizing: 1 and 2 have low GDD at 800 nm and reflectivities (transmissions) of 30 and 50% respectively; 3 is a low-reflectivity beam sampler; 4 is a broadband 750–1050 nm 50% reflector; and 5 is a 30% reflector and is followed by a 410 nm narrowband filter (6) to select pump SHG light used for Fabry–Pérot etalon locking. Half-wave plates (HWP) are used to prepare the pump and SC light into vertical polarizations before the waveguide. **b** Delay-resolved output from the MgO:PPLN waveguide, recorded by incrementing the temporal overlap between the pump and SC pulses using an optical delay line in the 800 nm pump channel. The offset of the delay axis is arbitrary, with the origin set to the delay which yielded the most intense SFM signal. Line cuts at 30 and 800 fs illustrate the enhanced spectral coverage and intensity achieved from the SFM process, spanning a spectral gap from 400–430 nm. **c** Photograph of the UV to blue-green supercontinuum generated by the MgO:PPLN waveguide at optimal delay between the pump and supercontinuum pulses. **d** Spectra of the fundamental pulses (red shading) and the SHG/SFM field (blue shading) generated in the MgO:PPLN waveguide. The shape and bandwidth of the SHG/SFM spectrum after Fabry–Pérot filtering (solid blue line) is preserved.

Further opportunities exist for developing the approach we describe here to better match it to the specific requirements of astronomical spectrographs. As described by ref. 12, it is preferable to mitigate the astrocomb's significant spectral structure, which is typically characteristic of nonlinear supercontinuum generation, in order to equalize the signal level of all the comb lines on the spectrograph CCD. Doing so improves the ultimate calibration accuracy by avoiding CCD saturation, minimizing the effect of photon noise, and allowing the best use of the available dynamic range of the CCD. Active spectral flattening is commonly implemented by using a liquid-crystal spatial light modulator[31] situated in the Fourier plane of a zero-dispersion grating compressor[32–34]. A simple feedback algorithm is used to clamp the intensity of all spectral components at a chosen level by attenuating all that exceed the target intensity. This procedure effectively flattens the comb spectrum to the intensity of the weakest mode, and is acceptable so long as the weakest mode has enough intensity to be registered with a high signal-to-noise ratio on the CCD in a reasonable exposure time. From an independent camera calibration measurement, we estimate the power per comb mode at the extreme long-wavelength (470 nm) edge to be 100 pW or 270 million photons/s. For the 4 K × 4 K HARPS3 CCD detector, with 30 GHz comb modes from 380–690 nm, with an estimated system throughput of 10%, this implies a minimum flux of 2 Mphotons/s, which would exceed the well depth of the HARPS3 CCD after 200 ms of exposure. This gives us confidence that implementing an additional spectral flattening

procedure would maintain the comb intensity from 390–470 nm at a level sufficient for making practical calibration measurements on an astronomical spectrograph, considering that (if necessary) calibration frames could be acquired over many seconds to compensate for higher loss or lower detection sensitivity in a real system.

In addition to an active spectral flattening approach, the spectral shape of the comb can be manipulated by optimizing the design of the MgO:PPLN grating. The MgO:PPLN grating used in experiments was designed from a priori modeling (see Methods); however, the design could be refined by using experimental pulse measurement data describing the intensity and phase of the fundamental fields. In combination with more sophisticated domain-level optimization strategies such as simulated-annealing[35,36], this could enable broader, more intense, and flatter spectra to be obtained, in turn demanding less intensive spectral shaping[31] before the spectrograph.

## Methods

### Nonlinear optical frequency conversion in MgO:PPLN

The second-order nonlinearity in lithium niobate is responsible for the sum-frequency and second-harmonic generation in the MgO:PPLN waveguide, which proceeds using the technique of quasi-phasematching (QPM)[37]. When Maxwell's wave equation is modified to include a second-order nonlinear polarization term[38], its solution describes the periodic energy exchange between three co-propagating fields with wavevectors $k_3 > k_2 > k_1$ and a "wavevector mismatch"

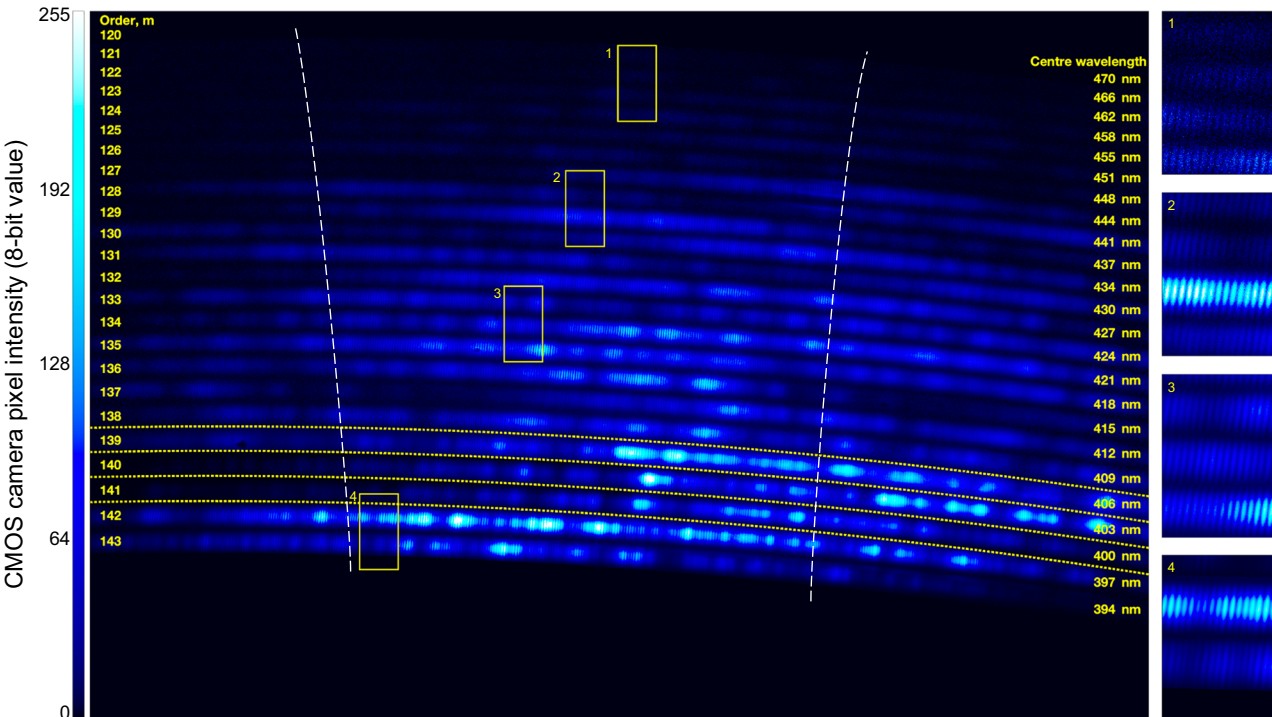

**Fig. 3 | Comb-mode-resolving echellogram.** Main image, a 20.2 megapixel camera frame recorded from a cross-dispersion spectrograph, with annotations indicating the index and center wavelength of each echelle order. The free spectral range of each order is 3–4 nm, equivalent to ~200 comb modes and indicated by the dashed white lines. Insets, magnified regions of the echellogram confirming that the 30 GHz comb modes are resolved across the full range of modes orders 120–143, corresponding to wavelengths from 392–472 nm. To compensate for the 8-bit dynamic range of the camera, orders 139–143, which are associated with the intense pump SHG light, were recorded at lower gains than orders 120–138. Regions bounded by yellow dashed lines share the same camera gain. In the main image, the color map is linear with intensity; a nonlinear color map has been used to enhance the visibility of the weaker comb modes in the insets.

defined as $\Delta k = k_3 - k_2 - k_1$. As the waves propagate, they accumulate a phase mismatch of $\phi_3 - \phi_2 - \phi_1 = \pi$ after one coherence length, $l_c = \pi/\Delta k$, by which point the maximum exchange of energy from fields 1 and 2 into field 3 has been achieved. Continued propagation in a bulk material would lead to back-conversion, but in QPM the structure of the material is periodically modulated so that the polarity of the second-order nonlinearity is reversed after each coherence length, resulting in a sustained transfer of energy into field 3. In practice, electrical-field poling[39] is used to create a periodically-poled (PP) longitudinal grating structure, in which the lithium-niobate crystalline domain direction alternates with a period of $\Lambda_G = 2l_c$ along the propagation direction. The wavelengths that are converted can be controlled by the choice of grating period, $\Lambda_G$, a physical parameter defined by the conductive lithographic electrode used for the poling process[39]. While simple devices typically employ a constant $\Lambda_G$, sophisticated control over the wavelength conversion properties can be obtained by spatially varying $\Lambda_G$ along the propagation direction. Analytic calculation[40], simulated-annealing[41] and genetic algorithms[42] can be used to design gratings where the length of each crystalline domain contributes to the realization of particular wavelength conversion outcome optimized for spectral shape and efficiency.

The design of our quasi-phasematched MgO:PPLN grating proceeded by combining an optimization algorithm with a forward model of the nonlinear conversion process[35,36], which we extended to broad bandwidths using a nonlinear envelope description[43–45]. The grating was designed entirely a priori, using a supercontinuum field first computed using a generalized nonlinear Schrödinger equation (GNLSE) model[46], whose critical parameters (pulse peak power; fiber nonlinearity) were pre-calibrated experimentally. The input pump pulse to the GNLSE model was a transform-limited $\text{sech}^2(t)$ pulse with a full-width at a half-maximum duration of 30 fs, a center wavelength of

800 nm, and an energy of 300 pJ, corresponding to an average power of 300 mW at a repetition frequency of 1 GHz. An identical pump pulse was used in the nonlinear envelope model for the SFM process.

To limit the search space, the local grating period (Fig. 1c) was parameterized as $\Lambda = \Lambda_1 + (\Lambda_2 - \Lambda_1)(z/l)^\alpha$, where the initial and final grating periods are $\Lambda_1$ and $\Lambda_2$ respectively, $z$ is the propagation distance along a waveguide of total length $l$, and $\alpha$ describes deviations from linear chirp. Various optimization strategies are possible, but to maximize the gap-free visible bandwidth we defined a merit function proportional to the total wavelength bandwidth exceeding a minimum threshold in spectral intensity. Simple phasematching calculations using the refractive-index equations for MgO:PPLN[47] were used to determine initial choices for $\Lambda_1$ and $\Lambda_2$ that bracketed the values corresponding to SHG and SFM of the participating fundamental wavelengths. The grating chirp parameter, $\alpha$, and the pump-supercontinuum inter-pulse delay, $\tau$, were initiated at values which were found manually to yield some SHG and SFM over the desired bandwidth. Optimization was performed in MATLAB® using the Nelder-Mead algorithm. The resulting grating design ($\Lambda_1 = 6.3\,\mu m$; $\Lambda_2 = 2.2\,\mu m$; $\alpha = 0.48$; $l = 5\,mm$) is shown in Fig. 1c, where light enters the facet of the waveguide containing the longest domain periods. Physically, the grating design can be understood as follows. For a target design yielding UV to green wavelengths from 380–515 nm—a range that comfortably exceeds our Fabry-Perot reflectivity bandwidth of 390–470 nm—the shortest wavelengths are produced by SHG of pump light from 760–840 nm in gratings with $\Lambda = 2.2$–$3.2\,\mu m$. Similarly, SHG of the most intense longer-wavelength supercontinuum components above 950 nm is phasematched by the domain periods from $\Lambda = 4.8$–$6.3\,\mu m$. Intermediate grating periods from $\Lambda = 3.2$–$4.8\,\mu m$ phasematch SFM between the intense 800 nm pump pulse and supercontinuum components from 890–1140 nm. The value of

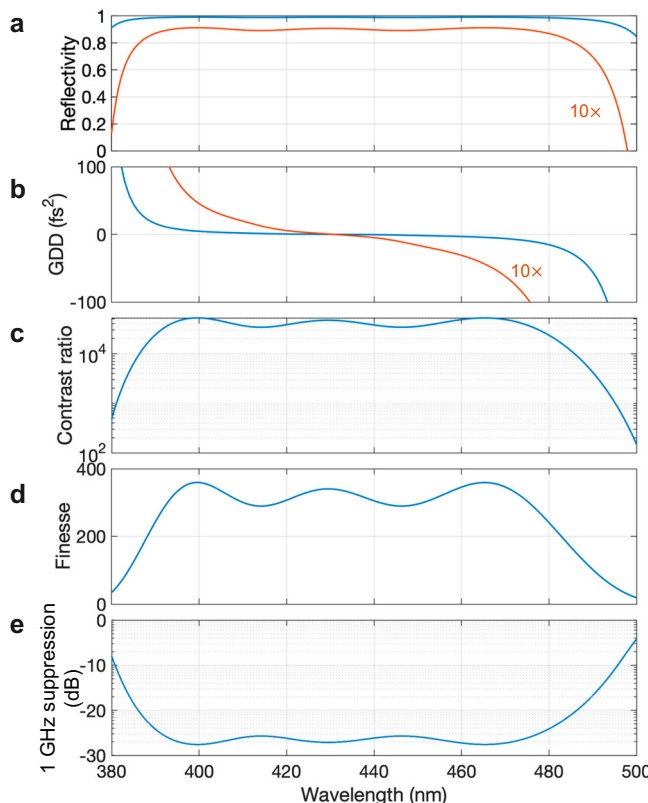

**Fig. 4 | Fabry–Pérot etalon performance. a** Wavelength dependent mirror reflectivity, with coatings having a nominal 99% reflectivity from 390–470 nm. **b** Group-delay dispersion (GDD) of the mirror coating. **c** Contrast ratio of the Fabry–Pérot etalon, defined as the ratio of maximum to minimum transmission. **d** Ideal Fabry–Pérot etalon finesse. **e**, Nearest-neighbor mode suppression.

$\alpha = 0.48$, determined by the optimization process, biases the design towards short and intermediate grating periods, compensating for the higher power in the supercontinuum at longer wavelengths. Supercontinuum wavelengths shorter than the pump wavelength are not converted by the grating.

## MgO:PPLN waveguide design and fabrication

In comparison to a bulk crystal, where diffraction limits the optical intensities which can be sustained for significant lengths, nonlinear frequency conversion in a quasi-phasematched lithium-niobate waveguide offers orders-of-magnitude higher conversion efficiency. Readily available commercial devices are typically weakly confining, with few-µm mode-field diameters being achieved by proton exchange[48] or mechanical dicing[19]. These devices are characterized by low coupling losses (≈0.5 dB), high damage thresholds[29], and dispersion properties similar to those of the bulk material, with second-order nonlinear effects typically dominant. More recently developed strongly confining thin-film PPLN devices have demonstrated direct multi-octave-spanning spectra extending into the blue and near-ultraviolet[49,50], and their conversion properties can be manipulated by dispersion engineering[51], including to optimize for UV generation[52]. Recently, a UV-to-green astrocomb with 18-GHz mode spacing has been reported using thin-film PPLN[53], derived from third- and fourth-order harmonic generation of a 1560 nm pump laser, although with spectral gaps in the wavelength coverage.

The MgO:PPLN waveguide was manufactured by Covesion using an electric-field-poling process in 0.5-mm-thick MgO:LiNbO$_3$ to create 0.3-mm-wide aperiodic gratings, followed by process steps of zinc indiffusion and ductile dicing to form ridge waveguides as described in ref. 19. A device length of 5 mm was chosen to support efficient

operation, and ridge widths of 11–13 µm were included to enable optimization of the modal overlap and conversion efficiency. The waveguide facets were anti-reflection coated at 380–520 nm and 800–1200 nm to minimize loss and etalon effects. Pump and supercontinuum light were co-launched into the waveguide using a plano-convex lens, achieving a mode-field diameter of ~5 µm. This weakly-confining design, while delivering a lower nonlinear conversion efficiency than thin-film PPLN, operates at intensities that avoid parasitic $\chi^{(3)}$-mediated or high-harmonic $\chi^{(2)}$ processes that can lead to some wavebands containing multiple combs with different offset frequencies[53]. The SFG and SHG processes promoted by our QPM-engineered grating result in a combined comb with a single offset frequency.

## Fabry–Pérot etalon filtering and locking

The Fabry–Pérot etalon was a plane-plane configuration with a spacing of ≈5 mm, adjusted to precisely match $c/2mf_{rep}$, where $m = 30$ corresponds to an astrocomb spacing of 30 GHz. The mirror performance is shown in Fig. 4 and the coating design (LaserOptik GmbH) was optimized to achieve a very low group-delay dispersion (GDD) from 390–470 nm without the need to employ a pair of complementary-dispersion coatings. This design achieves an etalon finesse >300 from 393–475 nm (Fig. 4d) and nearest-neighbor mode suppression (Fig. 4e) of nearly −30 dB over the same range. The UV to blue-green light leaving the MgO:PPLN waveguide was prepared in a collimated beam with a diameter of 5 mm, which slightly overfilled the clear aperture of the etalon. We spectrally conditioned the light to match the etalon's operating wavelength range by reflecting it multiple times on replica etalon mirrors before coupling it into the etalon. By doing this, we ensured that the intensity envelope of the filtered spectrum was identical to that of the input (see Fig. 2d), and that all spectral components of the incident comb were strongly filtered.

To enable active stabilization, one of the Fabry–Pérot mirrors was mounted on a fast, short-displacement piezoelectric transducer (PZT) and the other on a slower, longer-displacement PZT. To selectively transmit one family of sub-comb modes spaced at $30f_{rep}$ (30 GHz), the étalon mirror spacing was first set to ≈5 mm, then a few-nm, 10 kHz dither signal was applied to the fast PZT while the slower PZT was scanned until transmission peaks were obtained on a silicon avalanche photodiode (Fig. 2a). The detected signal was demodulated at the dither frequency, low-pass filtered and conditioned by a servo amplifier to stabilize the etalon in resonance with the selected sub-comb. Since the Ti:sapphire SHG (400 nm) component of the UV to blue-green spectrum was strongest, this light was responsible for the detected signal, which conveniently allowed the etalon locking to be pre-configured without the supercontinuum input to the waveguide. The identical comb offset frequencies of the SHG and SFM light mean that when the etalon is configured to mode-filter the pump SHG light, the supercontinuum SHG and pump-supercontinuum SFM outputs are also automatically transmitted. The stabilization technique was robust, capable of remaining in lock for several hours, and benefited from not requiring an auxiliary single-frequency UV/blue laser pre-locked to a comb mode.

## Cross-dispersion echelle spectrograph

The spectrograph comprised a 31.6 lines mm$^{-1}$ echelle grating (Thorlabs GE2550-0363) which was used in a quasi-Littrow configuration[54] with $\alpha \approx \beta \approx 63°$ to maximize the diffraction efficiency across each order. The echelle grating was tilted to achieve a small out-of-plane angle[54] of $\gamma \approx 1.5°$, allowing the vertically-dispersed diffraction orders to be intercepted by an F2-glass prism used close to minimum deviation. The prism provided cross-dispersion in the horizontal plane, following which a concave aluminum mirror with a focal length of 200 mm was used to image the echellogram onto a 20.2 megapixel 8-bit CMOS camera. The CCD sensors used in astronomical

spectrographs have much greater dynamic range (typically at least 16 bits) and lower dark noise than the sensor available to us. On such a spectrograph, the dynamic range of our astrocomb would be, in principle, fully resolvable, however strong saturation by the pump SHG light was avoided by reducing the camera gain. The associated echellogram orders (139–143) were extracted from a set of low-gain camera frames before being integrated into the main image shown in Fig. 3.

The resolving power ($R = \lambda/\Delta\lambda$) of a diffraction-grating spectrograph is $mN$, where $m$ is the diffraction order and $N$ is the number of illuminated grating elements. At $\alpha \approx 63°$, the 5 mm incident beam diameter illuminates 348 grating elements, corresponding to $R \approx 46{,}000$ at $\lambda = 420$ nm ($m = 134$). Expressed in frequency, this results in a spectral resolution of 15 GHz, sufficient to resolve individual comb modes at 30 GHz spacing.

## Data availability
The processed data supporting this study are available from Heriot-Watt University's Research Portal. https://doi.org/10.17861/4bc12cb2-9403-49aa-a7a4-2f32e645c7f1 (2024).

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

## Acknowledgements
The authors gratefully acknowledge grant funding from the UK Science and Technologies Facilities Council (ST/Y001273/1, ST/S001328/1, ST/V000403/1, ST/X002306/1, ST/N002997/1, and ST/V000918/1); the Royal Academy of Engineering (RCSRF1718639 and RCSRF2223-1678); and the Knut and Alice Wallenberg Scholarship 2021.

## Author contributions
R.A.M, S.T., N.P., and D.T.R. conceived and designed the experiments; K.D., Y.S.C., and T.M. performed the experiments and prepared the results; L.D.W. and C.B.E.G. designed and fabricated the MgO:PPLN waveguide. All the authors contributed to the writing of the paper.

## Competing interests
L.D.W. and C.B.E.G. declare competing interests associated with research funding from and/or employment by Covesion Ltd.; patents and/or patent applications assigned to the University of Southampton and/or Covesion Ltd. that are relevant to the subject matter of this manuscript. The remaining authors declare no competing interests.
