## [Peer Review File · Nature Communications]

REVIEWER COMMENTS

Reviewer #1 (Remarks to the Author):

I have reviewed the manuscript "Continuous Ultraviolet to Blue-Green Astrocomb" by Cheng et al. and find it to be a solid step forward in the development of broad-band, particularly blue astrocombs. It presents a novel approach, using the aperiodically-poled MgO:PPLN waveguide to generate a "gap free" spectrum starting around 390nm. This development can help guide future works to produce blue and gap-free comb spectra that may be useful for astronomy. I think the work's novelty and demonstrated progress warrant publication in Nature Communications.

My comments are relatively minor. The main one relates to often-used claim that the spectrum is "gap free". I do not disagree with this claim, but I don't think it's very clear what is really meant by it. Some might claim (incorrectly), for example, that it means that there's no modes which are much weaker than others. Clearly, Fig. 3 reveals that to be true, and so some might argue that there are, indeed, gaps in the spectrum. In practice, of course, a spectral flattener of some description will need to be used to even out the spectrum for real astronomical use. That is, the weakest comb mode effectively defines the intensity the rest of the comb modes will have after "flattening" in this way. So, there are no "gaps" as long as the weakest mode has enough intensity to be registered with high signal-to-noise ratio on the CCD in a reasonable exposure time (where "reasonable" relates to practice in most observatories, e.g. ~1 minute). To avoid misunderstanding, I would recommend being clearer about this somewhere in the paper (or Methods).

Also on Fig. 3, I found the lower-intensity modes very difficult to see, which rather undermines the points the authors are trying to make. Perhaps a logarithmic intensity scaling could be used instead of linear, to lessen the contrast between the brightest and faintest lines? I also found it difficult to see and follow the yellow dashed lines which define the regions sharing the same camera gain. It is also noted that the colour map is linear "in the main image" but what about the insets?

Overall the text and presentation were excellent and I found the manuscript very easy to read and understand. I did not notice any typographical errors except that a comma should replace the full stop in the sentence "The prism provided cross-dispersion in the horizontal plane. following which a concave...".

Reviewer #2 (Remarks to the Author):

This manuscript by Cheng et al. reports on generating a 30 GHz mode-spacing astronomical comb that can provide spectral coverage from blue-green to the ultraviolet region without gaps. The characteristic spectra were obtained by generating SHGs and SFMs with both supercontinuum (SC) and pump pulses in a designed MgO:PPLN waveguide. The continuous, 1 GHz mode-spacing comb was modified by a dispersion-compensated Fabry-Perot filter resonator to 30 GHz mode-spacing, yielding a calculated sidemode suppression ratio approaching -30 dB. The astronomical combs thus constructed were observed by the echelle spectrograph with high resolution and over a wide wavelength range, with each mode separated from the others.

The presentation of the experimental data and the conclusions drawn from them are soundly done, and the claims made in the manuscript are unquestionable. However, some points need to be adequately explained, and there is room for improvement. Fig. 1, which presents the concept of the study, needs to be more explained in terms of clarity. These shortcomings are discussed in detail below. Although the past literature is adequately cited for the most part, an introduction on nonlinear conversion processes in MgO:PPLN waveguide or related systems, needs to be included as noted below,

In the field of astronomical combs, the generation of combs in the short wavelength region, as in this study, is accompanied by technical difficulties. The wavelength conversion from the near-infrared fundamental to shorter wavelengths is often complex. In addition, it can be challenging to achieve low dispersion over a wide wavelength range in the short wavelength region because the dispersion of the dielectric multilayer material complicates the dispersion design of the filter resonator. In this study, a Ti:sapphire comb with a repetition rate of 1 GHz was used as the fundamental wave. The MgO:PPLN waveguide completed the necessary nonlinear optical processes other than SC on a single chip, and the high repetition rate of the fundamental wave reduced the difficulty of designing the filter resonator. In addition to the spectral features claimed by the authors, the reviewer believes that the experimental features described above are essential practical aspects for long-term stability and compactness.

The idea of achieving spectral uniformity through SFM and demonstrating this idea is interesting and meaningful from the perspective of nonlinear optics in general. The features of this comb may be particularly significant in the context of absolute frequency measurements of light, where the power per mode is critical to the reduction of the statistical uncertainty of the measurement. In the research area of astrocombs, which this study claims to be, it certainly advances the field in generating light with spectral features that have never existed before. However, the wavelength bands in which astronomical spectrometers excel vary widely, and it is difficult to say that presented principle is what many related researchers would consider adopting. Instead, 15 years have passed since the first proposal of the astronomical combs, and research has shifted to practical issues such as how to ensure the long-term stability of the combs in an observatory environment and how to achieve uniformity of spectral intensity in addition to spectral continuity. Therefore, while the results of this study may indeed be important for some high-dispersion spectrograph configurations, the reviewer is negative as to whether it is expected to have a significant impact on the majority of researchers of astrocombs, rewriting conventional concepts.

Here are my detailed comments.

(1) The authors claim that one of this work's key results is obtaining SHGs of both SC and pump pulses and their SFMs by a single MgO:PPLN waveguide. It is desirable to introduce what kind of studies there have been in the past regarding the generation of femtosecond pulses in the short wavelength region using related structures. This is expected to clarify whether this work applies an existing concept to astrocombs or a new way of utilizing a nonlinear medium.

(2) The design parameters of the MgO:PPLN waveguide are shown in Fig. 1c, but it would be beneficial to give a qualitative explanation of the relationship of this design to SHG and SFM. Also, the physical meaning of the equation in the figure should be explained.

(3) The spectral shape in Fig. 1d appears to be a simulation, but under what conditions was each component calculated? At least, it is different from the experimental results in Fig. 2.

(4) What kind of framework did the authors use to compute Figs. 1d and 1e? If the intent is to validate the idea, the reviewer needs to examine the validity of the simulations.

(5) Regarding Fig. 1e, there are wavelength regions where SFG and SHG have spectral overlap. In this case, the interference of the mutual electric fields may cause spectral modulation of the corresponding region. For example, do nonlinear phases in the SC generation or wavelength-scale changes in the relative delays between the SC and the pump not affect this interference effect? This is related to the long-term stability of the astronomical comb.

(6) In Fig. 2a, beam splitting and combining were carried out using cube-shaped beam splitters. Are these splitters polarizing beamsplitters? A description of each optical element is necessary in the caption. Also, the authors should explain how the polarization states of SC and pump were set when they were coupled to the waveguide.

(7) The authors should explain how they measured the ultrafast time-resolved spectrum shown in Fig. 2b. Did they use a streak camera? Or, is this the spectrum taken as a function of the delay between the pump and SC pulses? Also, please indicate how the time origin was defined.

(8) The need for the SFM process, which is the central concept of this manuscript, can also be discussed by showing the SHG spectra of SC pulses alone in the experiments, not in the simulations. Furthermore,

by showing the SHG spectrum of the pump pulse alone, one can discuss where the spectral gaps to be filled are and whether the SFM process actually fills them. The reviewer needs to confirm the consistency of the motivation for this study since the data in Fig. 2b seem to show no spectral gaps without the SFM process. In practical applications, the detectors have good sensitivities to detect weak stellar light, so even weak spectral components do not pose a severe problem.

Response to reviewers – Nature Communications manuscript NCOMMS-23-33868-T

The authors sincerely thank both reviewers for their careful comments, as a result of which we have made a number of careful revisions (detailed here), some based on further experimental measurements.

We believe the revised manuscript now sufficiently strengthens the arguments and claims contained in the paper to make it suitable for publication.

Our response (black) to comments from each reviewer (red) follows below, along with the consequent changes made to the manuscript text (blue).

Reviewer 1

We thank the reviewer for their very constructive comments, and note their support for the publication of this research. We address their comments below.

the weakest comb mode effectively defines the intensity the rest of the comb modes will have after "flattening" in this way. So, there are no "gaps" as long as the weakest most has enough intensity to be registered with high signal-to-noise ratio on the CCD in a reasonable exposure time (where "reasonable" relates to practice in most observatories, e.g. ~1 minute). To avoid misunderstanding, I would recommend being clearer about this somewhere in the paper (or Methods).

We appreciate the reviewer's constructive comments on this point. In response, we have:

(a) Introduced a significant new discussion on spectral flattening, including the addition of new references to the appropriate prior art [27–31]:

Further opportunities exist for developing the approach we describe here to better match it to the specific requirements of astronomical spectrographs. As described by Probst et al. [27], it is preferable to mitigate the astrocomb's significant spectral structure, which is typically characteristic of nonlinear supercontinuum generation, in order to equalize the signal level of all the comb lines on the spectrograph CCD. Doing so improves the ultimate calibration accuracy by avoiding CCD saturation, minimizing the effect of photon noise and allowing the best use of the available dynamic range of the CCD. Active spectral flattening is commonly implemented by using a liquid-crystal spatial light modulator [28] situated in the Fourier plane of a zero-dispersion grating compressor [29-31]. A simple feedback algorithm is used to clamp the intensity of all spectral components at a chosen level by attenuating all that exceed the target intensity.

(b) Added a calculation on photon flux which we consider provides an important insight into the applicability of our comb for practical calibration observations on typical astrophysical spectrographs:

This procedure effectively flattens the comb spectrum to the intensity of the weakest mode, and is acceptable so long as the weakest most has enough intensity to be registered with a high signal-to-noise ratio on the CCD in a reasonable exposure time. From an independent camera calibration measurement, we estimate the power per comb mode at the extreme long-wavelength (470 nm) edge to be 100 pW or 270 million photons / second. For the 4K × 4K HARPS3 CCD detector, with 30 GHz comb modes from 380–690 nm, with an estimated system throughput of 10%, this implies a minimum flux of 2 Mphotons / second, which would exceed the well depth of the HARPS3 CCD after 200 ms of exposure. This gives us confidence that implementing an additional spectral flattening procedure would maintain the comb intensity at a level sufficient for making practical calibration measurements on an astronomical spectrograph.

Also on Fig. 3, I found the lower-intensity modes very difficult to see, which rather undermines the points the authors are trying to make. Perhaps a logarithmic intensity scaling could be used instead of linear, to lessen the contrast between the brightest and faintest lines? I also found it difficult to see and follow the yellow dashed lines which define the regions sharing the same camera gain. It is also noted that the colour map is linear "in the main image" but what about the insets?

The colour maps used have already been adjusted to show the comb modes as clearly as possible. The insets already used a nonlinear scale to enhance comb-mode visibility and we have now made this clear in the caption to Fig. 3:

In the main image, the color map is linear with intensity; a nonlinear color map has been used to enhance the visibility of the weaker comb modes in the insets.

We have boldened the order-delineating lines and the adjacent text for clarity. For example:

Overall the text and presentation were excellent and I found the manuscript very easy to read and understand. I did not notice any typographical errors except that a comma should replace the full stop in the sentence "The prism provided cross-dispersion in the horizontal plane. following which a concave...".

We thank the reviewer for highlighting this; we have corrected it in the revised manuscript.

Reviewer 2

Although the past literature is adequately cited for the most part, an introduction on nonlinear conversion processes in MgO:PPLN waveguide or related systems, needs to be included as noted below.

We have responded to the reviewer's suggestion by substantially widening the scope of the first Methods section.

We have renamed this section, "Nonlinear optical frequency conversion in MgO:PPLN" and preceded the original text with an orientation paragraph for non-expert readers which first explains the fundamental principles of nonlinear frequency conversion in periodically-poled material.

Further, we support this section with the addition of appropriate references:

Nonlinear optical frequency conversion in MgO:PPLN

The second-order nonlinearity in lithium niobate is responsible for the sum-frequency and second-harmonic generation in the MgO:PPLN waveguide, which proceeds using the technique of quasi-phases-matching (QPM) [34]. When Maxwell's wave equation is modified to include a second-order nonlinear polarization term [35], its solution describes the periodic energy exchange between three co-propagating fields with wavevectors $k_3 > k_2 > k_1$ and a 'wavevector mismatch' defined as $\Delta k = k_3 - k_2 - k_1$. As the waves propagate, they accumulate a phase mismatch of $\phi_3 - \phi_2 - \phi_1 = \pi$ after one coherence length, $l_c = \pi/\Delta k$, by which point the maximum exchange of energy from fields 1 and 2 into field 3 has been achieved. Continued propagation in a bulk material would lead to back-conversion, but in QPM the structure of the material is periodically modulated so that the polarity of the second-order nonlinearity is reversed after each coherence length, resulting in a sustained transfer of energy into field 3. In practice, electrical-field poling [36] is used to create a periodically-poled (PP) longitudinal grating structure, in which the lithium-niobate crystalline domain direction alternates with a period of $\Lambda_G = 2l_c$ along the propagation direction. The wavelengths that are converted can be controlled by the choice of grating period, Λ_G , a physical parameter defined by the conductive lithographic electrode used for the poling process [36]. While simple devices typically employ a constant Λ_G , sophisticated control over the wavelength conversion properties can be obtained by spatially varying Λ_G along the propagation direction. Analytic calculation [37], simulated-annealing [38] and genetic algorithms [39] can be used to design gratings where the length of each crystalline domain contributes to the realization of particular wavelength conversion outcome optimized for spectral shape and efficiency.

In the research area of astrocombs, which this study claims to be, it certainly advances the field in generating light with spectral features that have never existed before. However, the wavelength bands in which astronomical spectrometers excel vary widely, and it is difficult to say that presented principle is what many related researchers would consider adopting. ...Therefore, while the results of this study may indeed be important for some high-dispersion spectrograph configurations, the reviewer is negative as to whether it is expected to have a significant impact on the majority of researchers of astrocombs, rewriting conventional concepts.

The reviewer is correct in stating that the potential impact for our work will be restricted to astronomical spectrographs with the highest resolution—but this is tautological, since low resolution spectrographs do not need the calibration precision, accuracy and traceability that astrocombs provide; they are sufficiently served by conventional hollow-cathode lamps. Therefore, all astrocomb technology (given its greater cost) is expected to be adopted primarily by the highest resolution instruments.

The reviewer states, our work "advances the field in generating light with spectral features that have never existed before", and indeed this is the key point. In order to better explain the significance of astrocomb calibration in the UV-blue/green spectral region, we have re-written the introduction to the article (see below), explaining the motivation for a frequency-comb calibrator here, based on the much higher density of emission lines in this spectral region. This is a strong argument for the need for astrocombs in this waveband, which until our work simply did not exist with sufficient bandwidth and / or continuous coverage to be considered a solved problem.

In the field of extreme precision radial velocity (EPRV) measurements, the aim is to push the detection threshold to 10 cm s^{-1} . This 10 cm s^{-1} goal is set to enable the detection of low-mass, longer period extra-solar planets (exoplanets), in particular those that are similar to our own Earth. On some of the next generation instruments, such as ANDES on the ELT [1], the aim is to achieve measurements at the $1\text{-}2 \text{ cm s}^{-1}$ level, to enable the high precision and high accuracy cosmological observations like the Sandage-Loeb Test [2].

To detect an exoplanet via the radial velocity (RV) method requires the collection of high-resolution spectra of its host star, over a timescale corresponding to at least three orbits of the exoplanet. The shift in wavelength of all the spectral lines is measured (Doppler shift) and recorded over time. For example, for the Earth in orbit around the Sun (viewed edge-on), this radial velocity timeseries would appear as a sinusoid with an amplitude of 9 cm s^{-1} and a period of one year. To put a 10 cm s^{-1} Doppler shift in perspective, the size of a single pixel on many current high resolution spectrographs used for RV measurements is of the order of 1 km s^{-1} , illustrating how these measurements require extremely precise wavelength calibration.

Astrocombs – broadband laser frequency combs with multi-GHz spacing – represent the ideal wavelength calibrator [3] for these tasks. To enable the detection of an Earth-analogue exoplanet, the RV measurement must be stable over many years, have a single measurement precision of $\approx 1 \text{ cm s}^{-1}$, have spectral features that are regular and sufficiently spaced, unresolved (i.e. narrow intrinsic width) and cover the entire spectral format so that maximum information can be extracted from the stellar spectrum. Laser frequency combs can provide all these features, by virtue of their stability, atomically-referenceable accuracy, sub-100-kHz linewidths, multi-GHz spacing and potential for broadband spectral coverage [4,5].

Radial velocity precision depends on the number of strong spectral features [6], and for solar-like stars the density of such lines increases towards the blue end of the spectrum. This is why the most advanced RV instruments today, e.g. HARPS (380–690 nm, [7]), EXPRES (380–680 nm, [8]) and ESPRESSO (378–789 nm, [9]), reach out to the blue part of the spectrum. Future science cases such as the Sandage-Loeb test use the cosmological redshifted Lyman-alpha forest, with the wavelengths starting at $\approx 365 \text{ nm}$. Converting this advantage to RV precision requires accurate and reproducible wavelength calibration in the blue and near-ultraviolet. Alternative calibration methods (principally hollow-cathode lamps) do not provide intrinsic accuracy, homogeneous wavelength coverage or stability of an LFC [3].

In relation to the question of "conventional concepts", there are so far none that adequately cover this waveband. Indeed, in general the astrocomb research community has not yet settled on an architecture of choice, with two major approaches being developed concurrently, and characterised by the sequence in which supercontinuum continuum generation and Fabry-Perot filtering are implemented.

(1) The authors claim that one of this work's key results is obtaining SHGs of both SC and pump pulses and their SFMs by a single MgO:PPLN waveguide. It is desirable to introduce what kind of studies there have been in the past regarding the generation of femtosecond pulses in the short wavelength region using related structures. This is expected to clarify whether this work applies an existing concept to astrocombs or a new way of utilizing a nonlinear medium.

We have re-written the Methods section "MgO:PPLN waveguide design and fabrication" to include a synopsis of PPLN waveguide devices for short-wavelength generation, including references to 6 additional research papers in this scope [49–54]:

MgO:PPLN waveguide design and fabrication

In comparison to a bulk crystal, where diffraction limits the optical intensities which can be sustained for significant lengths, nonlinear frequency conversion in a quasi-phased-matched lithium-niobate waveguide offers orders-of-magnitude higher conversion efficiency. Readily available commercial devices are typically weakly confining, with few- μm mode-field diameters being achieved by proton exchange [49] or mechanical dicing [19]. These devices are characterized by low coupling losses ($\approx 0.5 \text{ dB}$), high damage thresholds [29] and dispersion properties similar to those of the bulk material, with second-order nonlinear effects typically dominant. More recently developed strongly confining thin-film PPLN devices have demonstrated direct multi-octave-spanning spectra extending into the blue and near-ultraviolet [50,51], and their conversion properties can be manipulated by dispersion

engineering [52], including to optimize for UV generation [53]. Recently, a UV-to-green astrocomb with 18-GHz mode spacing has been reported using thin-film PPLN [54], derived from third- and fourth-order harmonic generation of a 1560 nm pump laser, although with spectral gaps in the wavelength coverage.

The MgO:PPLN waveguide was manufactured by Covision using an electric-field-poling process in 0.5-mm-thick MgO:LiNbO₃ to create 0.3-mm-wide aperiodic gratings, followed by process steps of zinc indiffusion and ductile dicing to form ridge waveguides as described in [19]. A device length of 5 mm was chosen to support efficient operation, and ridge widths of 11–13 μm were included to enable optimization of the modal overlap and conversion efficiency. The waveguide facets were anti-reflection coated at 380–520 nm and 800–1200 nm to minimize loss and etalon effects. Pump and supercontinuum light were co-launched into the waveguide using a plano-convex lens, achieving a mode field diameter of approximately 5 μm. This weakly-confining design, while delivering a lower nonlinear conversion efficiency than thin-film PPLN, operates at intensities that avoid parasitic $\chi^{(3)}$ -mediated or high-harmonic $\chi^{(2)}$ processes that can lead to some wavebands containing multiple combs with different offset frequencies [54]. The SFG and SHG processes promoted by our QPM-engineered grating result in a combined comb with a single offset frequency.

(2) The design parameters of the MgO:PPLN waveguide are shown in Fig. 1c, but it would be beneficial to give a qualitative explanation of the relationship of this design to SHG and SFM. Also, the physical meaning of the equation in the figure should be explained.

In Fig. 1c we have added the following text to the caption to better explain the actual grating dimensions and the physical meaning:

c, The MgO:PPLN domain-width pattern follows an aperiodic design optimized to ensure gap-free upconversion. The optimized QPM grating design has entry and exit periods of $\Lambda_1 = 6.3 \mu\text{m}$ and $\Lambda_2 = 2.2 \mu\text{m}$ respectively, and a nonlinearity of $\alpha = 0.48$, corresponding to a structure **(b)** with domain sizes that become progressively smaller towards the end of the waveguide.

In the Methods section "Nonlinear optical frequency conversion in MgO:PPLN" we have also added the numerical values for the equation and augmented the text to include a physical interpretation of the grating function:

The resulting grating design ($\Lambda_1 = 6.3 \mu\text{m}$; $\Lambda_2 = 2.2 \mu\text{m}$; $\alpha = 0.48$; $l = 5 \text{ mm}$) is shown in Fig. 1c, where light enters the facet of the waveguide containing the longest domain periods. Physically, the grating design can be understood as follows. For a target design yielding UV to green wavelengths from 380–515 nm—a range that comfortably exceeds our Fabry-Perot reflectivity bandwidth of 390–470 nm—the shortest wavelengths are produced by SHG of pump light from 760–840 nm in gratings with $\Lambda = 2.2\text{--}3.2 \mu\text{m}$. Similarly, SHG of the most intense longer-wavelength supercontinuum components above 950 nm is phasematched by the domain periods from $\Lambda = 4.8\text{--}6.3 \mu\text{m}$. Intermediate grating periods from $\Lambda = 3.2\text{--}4.8 \mu\text{m}$ phasematch SFM between the intense 800 nm pump pulse and supercontinuum components from 890–1140 nm. The value of $\alpha = 0.48$, determined by the optimization process, biases the design towards short and intermediate grating periods, compensating for the higher power in the supercontinuum at longer wavelengths. Supercontinuum wavelengths shorter than the pump wavelength are not converted by the grating.

(3) The spectral shape in Fig. 1d appears to be a simulation, but under what conditions was each component calculated? At least, it is different from the experimental results in Fig. 2.

We have now clarified this in the Methods section "Nonlinear optical frequency conversion in MgO:PPLN" with the following additional text:

The input pump pulse to the GNLSE model was a transform-limited $\text{sech}^2(t)$ pulse with a full-width at half-maximum duration of 30 fs, a centre wavelength of 800 nm and an energy of 300 pJ, corresponding to an average power of 300 mW at a repetition frequency of 1 GHz. An identical pump pulse was used in the nonlinear envelope model for the SFM process.

(4) What kind of framework did the authors use to compute Figs. 1d and 1e? If the intent is to validate the idea, the reviewer needs to examine the validity of the simulations.

As detailed in the Methods section "Nonlinear optical frequency conversion in MgO:PPLN", the simulation was produced by applying the theoretical framework originated by Conforti *et al.* [44]. This model is fully described in [45] and [46], where it has already been validated against broadband experimental data.

(5) Regarding Fig. 1e, there are wavelength regions where SFG and SHG have spectral overlap. In this case, the interference of the mutual electric fields may cause spectral modulation of the corresponding region. For example, do nonlinear phases in the SC generation or wavelength-scale changes in the relative delays between the SC and the pump not affect this interference effect? This is related to the long-term stability of the astronomical comb.

In principle, this is correct, and close inspection of Fig. 1e shows very weak temporal fringes in the regions where the SHG and SFG overlap. Note that the observed temporal period is not λ/c because the computational delay step size of 10 fs is insufficient to resolve this, leading to aliasing. However, in practice, the modulation depth of these fringes is weak.

We encountered no experimental issues associated with this effect, either when generating the astrocomb by Fabry-Perot filtering or when observing it on the high-resolution spectrograph. The effect is further mitigated experimentally by differences in the chirp between the SFG and SHG fields, which restrict any interference to a small delay range where identical frequencies are temporally coincident. Finally, we note that a basic stabilization loop, or even simply a kHz-rate few-100-nm dither of the delay, would be sufficient to fully mitigate this effect for astronomical observations.

(6) In Fig. 2a, beam splitting and combining were carried out using cube-shaped beam splitters. Are these splitters polarizing beamsplitters? A description of each optical element is necessary in the caption. Also, the authors should explain how the polarization states of SC and pump were set when they were coupled to the waveguide.

We have modified Fig. 2a to correctly show the beamsplitter types, and added labelling of this and other components:

Additionally, we have significantly extended the caption for Fig. 2 to more fully explain the system

a, Pump pulses from a 1 GHz Ti:sapphire comb enter a 35 mm long photonic crystal fiber to create 650–1150 nm supercontinuum (SC) pulses, which are launched into an MgO:PPLN waveguide (**inset**). Replica pump pulses are co-launched into the waveguide and overlapped with infrared SC components to achieve a broadband SHG and SFM comb, which is filtered by a Fabry-Pérot etalon to produce a 30 GHz astrocomb. All beamsplitters are non-polarizing: 1 and 2 have low GDD at 800 nm and reflectivities (transmissions) of 30% and 50% respectively; 3 is a low-reflectivity beam sampler; 4 is a broadband 750–1050 nm 50% reflector; and 5 is a 30% reflector and is followed by a 410 nm narrowband filter (6) to select pump SHG light used for Fabry-Pérot etalon locking. Half-wave plates (HWP) are used to prepare the pump and SC light into vertical polarizations before the waveguide.

(7) The authors should explain how they measured the ultrafast time-resolved spectrum shown in Fig. 2b. Did they use a streak camera? Or, is this the spectrum taken as a function of the delay between the pump and SC pulses? Also, please indicate how the time origin was defined.

We have clarified these details in the caption of Fig. 2 by adding the following text:

b, Delay-resolved output from the MgO:PPLN waveguide, recorded by incrementing the temporal overlap between the pump and SC pulses using an optical delay line in the 800 nm pump channel. The offset of the delay axis is arbitrary, with the origin set to the delay which yielded the most intense SFM signal.

(8) The need for the SFM process, which is the central concept of this manuscript, can also be discussed by showing the SHG spectra of SC pulses alone in the experiments, not in the simulations. Furthermore, by showing the SHG spectrum of the pump pulse alone, one can discuss where the spectral gaps to be filled are and whether the SFM process actually fills them. The reviewer needs to confirm the consistency of the motivation for this study since the data in Fig. 2b seem to show no spectral gaps without the SFM process. In practical applications, the detectors have good sensitivities to detect weak stellar light, so even weak spectral components do not pose a severe problem.

For consistency with the existing Fig. 2b, we have now included two new line cuts through the data. The first, at a large delay (800 fs), shows the SHG spectra of the pump and SC pulses and contains no SFM light; a low-intensity gap is clearly visible in the region from 400–430 nm. The second, at a delay close to optimum pump-SC overlap (30 fs), shows how this gap is filled by the SFM light. While it is true that there is some weak light already in this region, the beneficial impact of SFM is unquestionable: there is a broadband +10 dB increase in intensity, which in an application context reduces the dynamic range and attenuation needed from a subsequent spectral flattening module.

Line cuts at 30 fs and 800 fs illustrate the enhanced spectral coverage and intensity achieved from the SFM process, spanning a spectral gap from 400–430 nm.

REVIEWERS' COMMENTS

Reviewer #1 (Remarks to the Author):

I have read the revised manuscript "Continuous Ultraviolet to Blue-Green Astrocomb" by Cheng et al., plus the authors' responses to my own queries and those of Reviewer #2.

I am happy with the authors' response to my own queries, and their modifications to the paper as a result. I have no further comments and recommend the paper for publication.

I do not agree with Reviewer #2's concern that the (lack of) impact in demonstrating astrocomb performance at very blue wavelengths ($< \sim 400\text{nm}$). For astrocombs, this has been - and remains - a major shortcoming and difficulty, as demonstrated in the ESPRESSO spectrograph on the Very Large Telescope in Chile: the instrument covers wavelengths down to $\sim 380\text{nm}$ but even the recently upgraded astrocomb "only" reaches down to $\sim 420\text{nm}$ (which is itself considered impressively blue). Methods for producing useful astrocomb devices which also produce such blue signal are very important to investigate, and widely report, and will indeed have an important impact on this field because there are so many astronomically important spectral features at such wavelengths (e.g. heavy metal lines in stellar spectra, highly redshifted hydrogen Lyman alpha absorption lines from the intergalactic medium etc.).

I only think that the text of the article should make it clear that there is enough photons in the lowest-intensity spectral ranges of the comb system such that, for any realistic train of optics that would be fed by the comb system, and any realistic detector efficiency, the signal-to-noise ratio of the detected comb light will be sufficient to use it for calibration purposes.

I think the authors' response to this criticism from Reviewer #2 is appropriate. Indeed, it is along similar lines to what I've outlined above. The modified text in the article is also fine, and makes the need for blue light clear enough.

Reviewer #2 (Remarks to the Author):

The authors have addressed appropriately many of the reviewer's comments, and the manuscript has improved significantly regarding readability and reproducibility. However, in response to the most critical opening comment of the reviewer that the wavelength bands in which observatories excel depend on

their spectrographs, the authors shifted the point to tautological and made a nonessential claim. It is a given that astronomical combs are used with high-dispersion spectrometers. What the reviewer stated was that the wavelength bands covered by some high-dispersion spectrometers are in the near-infrared region, some in the visible region. The reviewer reinforces the opinion that the present study does not necessarily have a significant impact on most astrocomb researchers. This is a study of a spectral enhancement of an astrocomb with a limited wavelength range, as evidenced by the title and the added description of several high-dispersion spectrographs operating at short wavelength regions in revised manuscript.

More importantly, the authors state in responding the reviewer's comment (8) that weak light is present even without SFM, just as the reviewer was concerned. Reviewer 1 also commented on the same point, but the author avoided mentioning the "gap" wavelength region. It is now clear that the motivation of this study is not to fill the real spectral gap but to contribute to reducing the dynamic range and light attenuation during subsequent spectral flattening. Essentially, spectral flattening is performed relative to weak light levels, and the need for SFM can only be justified by discussing logically how essential a +10 dB improvement is. It should be noted that this is by no means "unquestionable", which is why the reviewer offered comment (8). With spectral flattening, the SFM, which is the central claim of the present work, is unnecessary. In the "gap" region, the SHG already had a comparable intensity as that at 470 nm, as can be inferred from Fig. 2b and 2d in the revised manuscript.

In conclusion, some of the reviewer's essential concerns were not properly addressed. The reviewer was originally neutral but now cannot recommend publication.